# Burnout in Intensive Care Unit Workers during the Second Wave of the COVID-19 Pandemic: A Single Center Cross-Sectional Italian Study

**DOI:** 10.3390/ijerph18116102

**Published:** 2021-06-05

**Authors:** Nino Stocchetti, Giulia Segre, Elisa R. Zanier, Michele Zanetti, Rita Campi, Francesca Scarpellini, Antonio Clavenna, Maurizio Bonati

**Affiliations:** 1Neuroscience Intensive Care Unit, Department of Anesthesia and Critical Care, Fondazione IRCCS Ca’ Granda Ospedale Maggiore Policlinico, 20122 Milan, Italy; nino.stocchetti@policlinico.mi.it; 2Department of Pathophysiology and Transplantation, University of Milan, 20122 Milan, Italy; 3Laboratory for Mother and Child Health, Department of Public Health, Istituto di Ricerche Farmacologiche Mario Negri IRCCS, 20156 Milan, Italy; giulia.segre@marionegri.it (G.S.); michele.zanetti@marionegri.it (M.Z.); rita.campi@marionegri.it (R.C.); francesca.scarpellini@marionegri.it (F.S.); antonio.clavenna@marionegri.it (A.C.); 4Laboratory of Acute Brain Injury and Therapeutic Strategies, Department of Neuroscience, Istituto di Ricerche Farmacologiche Mario Negri IRCCS, 20156 Milan, Italy; elisa.zanier@marionegri.it

**Keywords:** intensive care units, burnout, psychological distress, COVID-19, health personnel

## Abstract

The COVID-19 pandemic had a massive impact on the Italian healthcare systems, which became overwhelmed, leading to an increased risk of psychological pressure on ICU workers. The present study aimed to investigate the prevalence of distress (anxiety, depression and insomnia symptoms), burnout syndrome and resilience in healthcare workers during the COVID-19 pandemic and to detect potential factors associated with their psychological response. This cross-sectional, survey-based study enrolled 136 healthcare workers assisting COVID-19 patients in the new COVID-19 ward (Intensive Care Unit), at Milano Fiera, Lombardy. Participants completed an online survey that comprised different validated and standardized questionnaires: Maslach Burnout Inventory (MBI), Resilience Scale for adults (RSA), Hospital Anxiety and Depression scale (HADS) and Insomnia Severity Index (ISI). Socio-demographic and work characteristics were also collected. Out of 136 ICU specialists, there were 84 nurses (62%) and 52 physicians (38%). Over half (60%) met the criteria for burnout, with nearly the same percentages among nurses and physicians. Nurses reported significantly higher scores of anxiety and insomnia levels. Forty-five percent of participants reported symptoms of depression (of whom 13.9% in the clinical range) and most of the staff showed moderate to high levels (82.4%) of resilience. The COVID-19 pandemic can have a significant impact on ICU staff. Effective interventions are needed to maintain healthcare professionals’ mental health and relieve burnout. Follow-up and tailored procedures should be provided to alleviate the psychological burden in the frontline staff at highest risk.

## 1. Introduction

Working in the Intensive care unit (ICU) is already a source of stress under normal circumstances. Data collected before the COVID-19 pandemic revealed that 13% of ICU professionals were anxious, 4% depressed, and 11% presented symptoms of post-traumatic stress disorder (PTSD) [1,2]. The risk of developing PTSD was greater in nurses working in ICUs than in other hospital wards [3].

A review conducted before the pandemic [4] found that the prevalence of burnout in ICU healthcare workers ranged from 6–47% and was associated with age, sex, marital status, personality traits, work experience in an ICU, work environment, workload and shift work, ethical issues, and end-of-life decision-making. The COVID-19 outbreak was declared a pandemic by the World Health Organization in March 2020. The spread and incidence of this new virus quickly forced a large part of the world’s countries to take drastic health measures to protect populations and health systems, which became overwhelmed. Italy was the first European country to implement lockdown measures. In Lombardy, the most affected Italian region, [5] a high mortality rate (26%) was found in COVID-19 patients admitted to ICU. The epidemic put significant strain on the regional healthcare system [6]: the hospitals’ organization was revised, COVID-19 wards were created ad hoc, and ICU beds were added. ICU specialists manage critically ill COVID-19 patients with high mortality rates and are exposed to psychological risk: fear of being a carrier of the virus and of putting their family, friends, and colleagues at risk, and fear of dying. Caregivers are particularly concerned about contracting the virus and spreading it to others [7].

The outbreak resulted in both physical and mental fatigue among physicians and nurses: the disproportion between the need for technological supplies in intensive care and their scarcity promotes, among many factors, higher levels of psychological stress. Anxiety, irritability, insomnia, fear, and anguish were observed during the pandemic, probably related to extremely high workloads and the lack of personal protective equipment [8]. In particular, the level of fear of ICU employees (nurses and physicians) was higher than that of hospital ward co-workers [9]. A recent survey [10] investigated mental health outcomes in British staff working in ICUs between June and July 2020. Over half (59%) reported good well-being, while 45% met the threshold for probable clinical significance for at least one of the following measures: severe depression (6%), PTSD (40%), severe anxiety (11%), or problem drinking (7%). During the peak of the “crisis period”, when the influx of patients was the highest (first phase of the pandemic) in Lyon, nearly half of the ICU caregivers reported significant levels of anxiety, while depression and PTSD incidence were 16% and 27%, respectively [11].

Several studies conducted during the first coronavirus wave showed that physicians and nurses experienced depression and anxiety as a result of the pandemic, which may have triggered burnout [12,13,14].

Burnout is a state of physical, emotional, and mental exhaustion resulting from long-term involvement in work situations that are emotionally demanding [15]. Burnout might be exacerbated by chronic, interpersonal, job-related stressors, especially for caregiving occupations such as healthcare [16]. This syndrome has three dimensions: emotional fatigue, depersonalization, and lack of personal accomplishment, and these are negatively associated with professionals’ mental and physical health. Burnout, in fact, predicts symptoms of depression, anxiety, PTSD, alcohol abuse, insomnia, and even suicide ideation [17].

Resilience might protect ICU workers from burnout, as reported in a Canadian qualitative study [18]. Resilience is described as the ability to face problems and adapt to adverse conditions [19,20]. Two recent studies explored the relation between resilience and burnout among healthcare workers during the COVID-19 pandemic: they revealed that psychological resilience could have a mediating role between mental health outcomes and all burnout dimensions [17,20].

Based on recent expertise during the pandemic [21,22], the survey was organized and managed according to previous methods used in survey studies by the authors. The present study aimed to investigate the psychological impact of the pandemic on front-line staff working at the new COVID-19 ward at Milano Fiera, Lombardy during the second pandemic wave. This temporary healthcare structure was built ad-hoc to cope with the demand for ICU beds for critically ill COVID-19 patients. We explored the prevalence and extent of distress, burnout syndrome, and resilience in ICU workers, and associated factors.

## 2. Materials and Methods

Data for this cross-sectional web-based study were collected from January 11th to 28th, 2021. We contacted ICU specialists through the hospital’s mailing list and HCW’s chats, sending them a brief explanation of the purpose of the study and a link for filling in the online questionnaire. A dedicated website (https://icufiera.marionegri.it/) was created ad hoc for this project. An online, semi-structured questionnaire was developed by using Wordpress, a free open-source content management system (CMS), integrated with SurveyJS (survey library and survey creator), a library to facilitate survey creation and management. The survey script was available for all devices.

Online consent was obtained from the participants. The survey questionnaire was anonymous and required approximately 15 minutes for completion. No participant identifier was required or recorded, preserving the anonymity of responders. Once data were extracted, they were checked to prevent duplicates; incomplete questionnaires were excluded from the Maslach Burnout Inventory. The ethics committee of the Besta Neurological Institute in Milan approved the study protocol (Report number 81). All the items of the STROBE checklist for cross sectional studies were met in the present report.

The first section comprised demographic questions and information regarding personal and professional experience; this section explored potential associated factors including age, gender, living conditions (people living with the participant), occupation, work hours, length of professional experience, and usual place of work (hospital unit before the pandemic). The second part of the survey focused on mental health outcomes: symptoms of burnout, anxiety and depression were investigated as well as resilience strategies.

The following validated assessment tools were used: the Maslach Burnout Inventory (MBI) [23] is a 22-item questionnaire asking respondents, on a 7-point Likert scale (from 0 to 6), the frequency with which they had recently experienced specific feelings related to their work. The MBI evaluates three subscale domains: emotional exhaustion (EE, measures feelings of being emotionally overextended and exhausted by one’s work; nine items), depersonalization (DP, measures unfeeling and impersonal response toward recipients of one’s service, care or treatment; five items), and personal accomplishment (PA, measures feelings of competence and successful achievement in one’s work with people; eight items). Participants were considered to have experienced burnout when the level of emotional exhaustion or depersonalization exceeded the cut-off, regardless of the presence or absence of reduced personal accomplishment, given its lack of evidence as a predictor of clinically diagnosed burnout [15,24]. Consistent with the Italian literature, we considered the level of burnout to be high if emotional exhaustion scores were ≥24, personal accomplishment scores were ≤29, and depersonalization scores were ≥9; moderate if emotional exhaustion scores were 15–23, personal accomplishment scores were 30–36, and depersonalization scores were 4–8; and low if emotional exhaustion scores were ≤14, personal accomplishment scores were ≥37, and depersonalization scores were ≤3 [25,26].

The Hospital Anxiety and Depression Scale (HADS) is a 14-item self-report screening scale originally developed to indicate the possible presence of anxiety and depression states in the setting of a medical, non-psychiatric outpatient clinic. The HADS consists of a 7-item anxiety subscale and a 7-item depression subscale. A score of >8 identifies subjects with a positive history for anxiety and/or depression [3,27,28]. For each scale, scores of 8–10 indicate mild symptoms (possible case) and a score greater than or equal to 11 indicates moderate/severe symptoms (probable case).

The Resilience Scale for Adult (RSA) [29,30] is a 14-item self-report instrument for evaluating six protective dimensions of resilience in adults. Item responses range from 1 (strongly disagree) to 7 (strongly agree) and scores vary between 14 and 98, with higher scores indicating higher levels of resilience. A score <56 indicates a very low resilience level; a score between 57 and 64 indicates a low resilience level; a score between 65 and 73 indicates that resilience level is on the low end; a score between 74 and 81 indicates a moderate resilience level; a score between 82 and 90 indicates a moderately high resilience level; and a score >91 indicates a high resilience level.

The Insomnia Severity Index (ISI) questionnaire asks respondents to rate the nature, severity, and impact of insomnia using a Likert-type scale. Questions relate to subjective qualities of the respondent’s sleep, including the severity of symptoms, the respondent’s satisfaction with his or her sleep patterns, the degree to which insomnia interferes with daily functioning, how noticeable the respondent feels his or her insomnia is to others, and the overall level of distress created by the sleep problem. Though developers point out that their chosen cutoff scores have not been validated, they offer a few guidelines for interpreting scale results: a total score of 0–7 indicates “no clinically significant insomnia”, 8–14 means “subthreshold insomnia”, 15–21 is “clinical insomnia (moderate severity)”, and 22–28 means “clinical insomnia (severe)”.

### Statistical Analyses

Data are reported as number and percentage of responders. Data analysis was performed using frequency distributions for categorical variables summarized using proportions and associations tested using chi-square or Fisher’s exact test (χ^2^ or F)**,** where applicable.) Continuous variables were tested for normality with Kolgorov-Smirnov (“age” was the only continuous variable that we included, D = 0.0119792, *p* < 0.01) and summarized using mean, median, and range.

To identify factors influencing burnout and the MBI scale, we computed odds ratios (OR) considering the significance of the confidence intervals (CI). Statistical significance was evaluated using 95% confidence interval and a two-tailed *p*-value of <0.05.

In the multivariable analysis, a log-binomial regression model was used. All variables were entered into the model and a stepwise regression analysis was conducted.

SAS software, version 9.4 (SAS, Institute Inc., Cary, NC, USA) was used for all statistical analyses.

## 3. Results

Of a total of 271 nurses and physicians working in this ICU, 150 (55% mean response rate; 71.2% for physicians, 42.4% for nurses) completed the survey questionnaire, of whom 14 (9%) were excluded because of missing data on the MBI. No difference was found between responders and non-responders for gender and profession (physicians vs. nurses).

The data on the 136 study participants (84 nurses and 52 doctors) were analyzed (Table 1). The variable “age” presented a continuous distribution, and the mean value of the participants was 39.1 years; most (74.1%) were living with others, and 44% had children.

The majority (67.4%) of the intensivists had already been working in ICUs before the pandemic, but nearly half (54%) reported difficulties in adapting to the new work environment (COVID-19 ICU). Nearly one out of three participants was not working in an ICU before the pandemic; in particular thirty nurses used to work in other hospital wards: this could have led to a higher percentage of reported difficulties in the ICU environment for nurses (59.7%).

The only statistically significant difference between nurses and physicians (*p* = 0.02) was observed in the fear of spreading the virus to people they lived with: nearly 70% of nurses was very worried vs. 56.7% of physicians.

ICU specialists were asked to compare their COVID-19 experience to their prior work. Nearly half of nurses did not report an increased workload, whereas two out of three physicians (68.8%) experienced it. Most nurses stated that the opportunities to discuss important decisions in group were less frequent than before, while physicians reported the opposite.

Most of the HCWs (58%) felt protected when working: the most common reported adverse reactions of wearing protectors (personal protective equipment, PPE) for long time were eye strain, thirst, and the impossibility of using the toilet.

Shifts were fixed: nurses had longer shifts (mean number of hours per shift) and a higher number of night shifts per month than physicians.

### 3.1. Mental Health Outcomes

The mean rate for the three dimensions of burnout is shown in Figure 1. A high level of burnout (60.3%) was frequent among the respondents (Figure 1a). High levels of depersonalization (DP) were observed in 47.8% of the intensivists (Figure 1b): more than half of the nurses reported high scores (54.8%), while lower scores were observed in the physicians (36.5%) (Table A1). A high level of emotional exhaustion (EE) was present in 41.2% of the respondents (Figure 1c) Low levels of personal accomplishment (PA) were found in 52.9% of the intensivists who responded to the survey (Figure 1d).

The proportion meeting the criterion for burnout in this study (high EE or DP) was 60.3% (82 participants), with nearly the same percentages among nurses (61.9%) and physicians (57.7%).

22.7% of participants received a score referring to a high risk of burnout, in the high range for all three; nearly the same prevalence of intensivists (22%) reported low levels in all the three dimensions.

Anxiety symptoms were reported by 53% of participants (Figure 2a), with a statistically significant difference (*p* = 0.02) between nurses and physicians: while the majority of nurses reported symptoms of anxiety (61.5%), most of the physicians presented normal levels of anxiety (60%).

Depression symptoms were found in 45.2% of responders (Figure 2b). In particular, more than one out of two intensivists (54.8%) reported normal levels of depression, with no differences between nurses and physicians. Overall, nearly one out of five (16.1%) scored above 11 for anxiety on the HADS, indicating the presence of clinical levels of anxiety, whereas 13.9% scored in this clinical range for depression.

Participants showed moderate/high levels (82.4%) of resilience (Figure 2c). Only 8.7% of the ICU staff was highly resilient, while 17.6% had low or very low scores. 61.5% of the intensivists described insomnia symptoms (Figure 2d), and there was a significant difference between nurses and physicians (*p* = 0.005): 71% of nurses reported symptoms of insomnia, while physicians showed lower percentages (54.3%), with clinically significant symptoms in only 17.6%.

### 3.2. Factors Associated with Burnout

From univariate analyses, factors associated with burnout were: anxiety, depression, and insomnia symptoms (Table A2).

The logistic regression analysis (Table 2) showed that working in other hospital wards (not ICU) before the pandemic is a possible predictor of lower levels of burnout (OR 3.02, 95% C.I. 1.06–8.58, *p* < 0.05). High levels of depressive symptoms were also found as a factor significantly associated with burnout (OR 4.88, 95% C.I. 1.54–15.48, *p* < 0.01).

Considering intensivists and burnout, high levels of depersonalization were found in 65 out of 82 participants, while a high degree of emotional exhaustion was reported in 56 intensivists.

A subgroup analysis was conducted for the three dimensions of burnout (Appendix A).

Presence of symptoms of anxiety, depression, and high levels of resilience were associated with high emotional exhaustion, high depersonalization, and lower levels of personal accomplishment. Clinical insomnia symptoms among intensivists were strongly associated with high levels of EE. (OR 4.71, 95% C.I. 2.04-10.84). Regarding gender, men reported lower levels of high EE than women (OR 0.46, 95% C.I. 0.22–0.94). Physicians had lower levels of DP than nurses (OR 0.48, 95% C.I. 0.23–0.97). High levels of burnout (high EE/ low PA) were reported in those who did not feel protected when working (respectively OR 2.94, 95% C.I. 1.41–6.09 for EE, and OR 2.98, 95% C.I. 1.44-6.16 for PA)

Difficulties in adapting to the new work environment were related (OR 2.34, 95% C.I. 1.11–4.96) to emotional exhaustion: those who described more difficulties were twice as likely to have high levels of burnout than those who did not report them.

## 4. Discussion

The results of the present study confirm that COVID-19 had a significant adverse impact on the psychological well-being of ICU workers.

A recent meta-analysis [31] examined the psychological burden of frontline medical staff during pandemics and epidemics and concluded that the prevalence of burnout symptoms was 31.8%.

In the present study, burnout levels were higher than those found in other studies: 43% of respondents scored at high risk for burnout on at least one dimension (high EE, high DP or low PA), 60.3% met the criterion for burnout (EE or DP), but only 22.7% were in the high range for all three. Regarding the three dimensions, high levels of reduced personal accomplishment were found in half of the intensivists, followed by high levels of depersonalization and emotional exhaustion scores (47.8% and 41.2%, respectively).

Our results are consistent with other studies conducted during the COVID-19 outbreak: a recent study reported that half of the European intensivists (51%) experienced severe burnout [13]. Similar rates (49.5%) of moderate to severe personal burnout were described in another study conducted in Singapore during the same period [32]. A higher prevalence of burnout (82.1%) was reported in China among ICU physicians [33].

Regarding the three dimensions of burnout, an Italian study highlighted higher scores for emotional exhaustion (40.7%) than depersonalization (30.2%) and low personal accomplishment (36.4%) [25]. Lower scores of burnout (high EE = 37%, high DP = 25%) were found in a recent Italian study [34]. A possible reason for this could be that the researchers considered HCW who directly assisted COVID-19 patients, but did not focus specifically on ICU workers. Average levels of emotional exhaustion and depersonalization were described in previous studies concerning Italian HCWs in normal working conditions [35], but the baseline levels of burnout depended on the subject characteristics. An Indian study showed that, during the emergency, nurses experienced moderate to severe levels of burnout in emotional exhaustion (54.2%) and depersonalization (43.3%), but mild to moderate levels of burn out in reduced personal accomplishment and presented a moderate to high level of resilience [36]. Similar rates of resilience were also reported in the present study (82.4% moderate or high resilience), with no significant differences between nurses and physicians.

Findings of one study [1] conducted before the outbreak, and only on ICU staff, showed that the prevalence of burnout was 37%: in contrast with our results, they found that physicians were twice as likely as nurses to be at risk of reporting burnout. Physician fatigue has a negative impact not only on one’s well-being but also on patient care and the health care system. This may be consistent with low job satisfaction, decreased work productivity, medical errors, poor quality of patient care, low job satisfaction, early retirement, and healthcare system failure [37]. Moreover, there was a degree of overlap between burnout and other measures of distress, most notably for anxiety.

Anxiety and depression symptoms were more common among nurses than doctors. Our findings are comparable to those of an Italian study [38] conducted on HCW during COVID-19 that demonstrated that staff working in ICUs or sub-intensive COVID-19 units had a significantly increased risk of developing adverse psychological outcomes (more specifically, post-traumatic distress symptoms and depression), independent of any other factor, and that nurses had a considerably greater risk of adverse psychological outcomes than physicians. One study conducted in the UK on ICU workers during the pandemic [39] showed that higher levels of anxiety and depression were reported in females and in nurses than physicians. Shen et al. [40] revealed that ICU nurses have to face many difficulties such as working in an unfamiliar environment, lack of experience in caring for infectious patients, anxiety about being infected, heavy workload, extreme exhaustion and depression, due to failure to treat critically ill patients. A recent qualitative study [41] drew attention on the challenges faced by nurses working in intensive care units during the crisis of the COVID-19 pandemic. The study reported that an organization’s inefficiency in supporting them, physical exhaustion, living with uncertainty, and the psychological burden of the disease were the most challenging themes associated with caring for COVID-19 patients.

In the current study, symptoms of insomnia were frequent among intensivists (61.5%). In particular, 17.6% of the participants reported clinically significant symptoms of insomnia: higher rates of moderate or severe insomnia were more frequent in nurses than in physicians (5.7%). High prevalence of insomnia symptoms was also found in a Chinese study conducted on front-line nurses fighting against COVID-19 [42], which reported that 52% of nurses showed insomnia. Comparing these results with our findings, it is essential to take into consideration that data were collected in different phases of the pandemic (March 2020, first phase) and assessed with another self-report measure (Ascension Insomnia Scale).

A positive relationship between exhaustion and insomnia was also observed [43]: frontline medical staff are under enormous pressure during the COVID-19 pandemic. Expectations leading to anxiety, depression, stress-related symptoms, insomnia, and worry about becoming infected or infecting family members can lead to exhaustion. Stress could be considered the primary cause of insomnia: a relationship between insomnia and other related psychological effects of working in hospitals was identified during the previous SARS outbreak [44]. Concerning resilience, results were not as expected. According to a recent study [17], resilience is a partial mediator in the relationship between emotional exhaustion and depersonalization with mental health outcomes. In the current study, moderate or high levels of resilience were reported in the majority of participants; our findings are in agreement with another recent study [45], which reported that 29% of physicians with the highest possible resilience score had burnout. This unexpected result could suggest that the level of resilience was not enough to prevent ICU specialists from developing burnout symptoms, or even that a high level of resilience was not enough to impede the development of extremely high level of burnout during COVID-19 pandemic. 

The relevance of our findings should be viewed considering a few limitations. Firstly, the sample was not representative of Italian ICU workers because we focused on ICU staff working in the Milano Fiera COVID-19 ward; the results are therefore specific to the involved area and may not necessarily be generalizable to other regions differently affected by the pandemic. Secondly, self-report measures were used to investigate psychological symptoms, and not a disorder that needs an appropriate diagnosis. Lastly, the cross-sectional design precluded causation and inferences: there is a lack of comparison with the pre-COVID-19 period and follow-up studies are needed to monitor mental health outcomes over the course of the pandemic and to assess long-lasting effects of psychological symptoms once the imminent threat of COVID-19 recovers. However, a recent study [14], assessed the prevalence of burnout symptoms in ICU professionals before and during the coronavirus disease 2019 crisis; their findings demonstrated that prevalence rose from 23% before COVID-19 to 36.1% at post-peak time, highlighting that overburdening of ICU professionals during an extended period of time leads to symptoms of burnout.

## 5. Conclusions

The COVID-19 pandemic had a negative effect on the mental health and psychological wellbeing of healthcare workers, in particular for those who were in strict contact with COVID-19 patients in the intensive care units. The large presence of burnout among HCWs, without differences between nurses and physicians, suggested that the current situation has influenced the level of distress of all stakeholders, increasing anxiety and depression symptoms and changing sleep rhythms. A protracted exposure to COVID-19 patients in the intensive care unit leads to a psychological burden in medical staff. This study demonstrated that resilience in uncertain, scarcely-supported situations is not enough. For this reason, awareness about healthcare workers’ psychological well-being and preventive interventions can positively influence the development of burnout syndrome. Follow-up and tailored procedures should be provided to alleviate the psychological burden to the frontline staff at highest risk. This study was conducted during the second pandemic wave, suggesting that burnout is a persistent risk for ICU workers.

## Figures and Tables

**Figure 1 ijerph-18-06102-f001:**
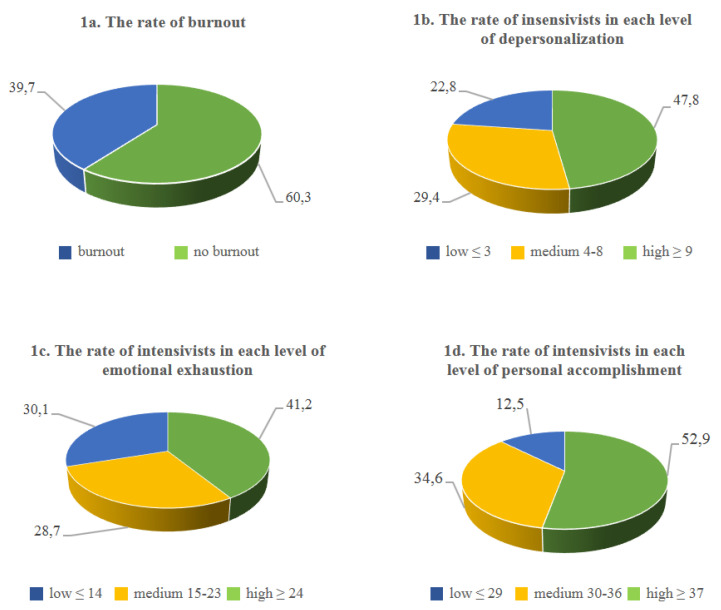
The rate of burnout and the rate of intensivists in each level of the three dimensions of burnout (variables are expressed as percentages).

**Figure 2 ijerph-18-06102-f002:**
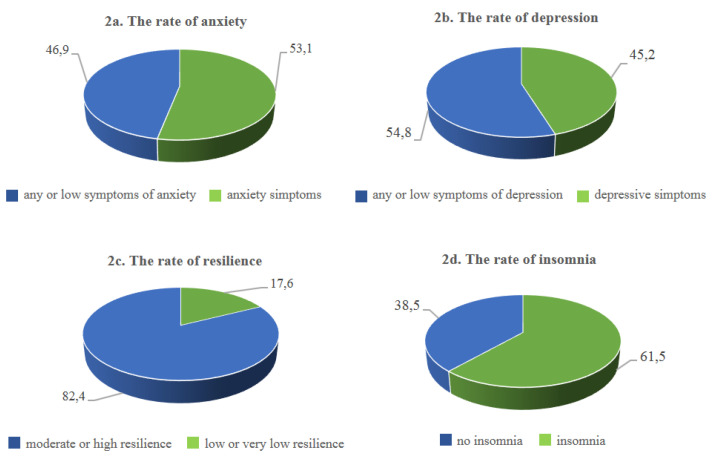
The rate of anxiety, depression, resilience and insomnia among intensivists (variables are expressed as percentages).

**Table 1 ijerph-18-06102-t001:** Demographic indicators.

	Nurses (N = 84)	%	Physicians (N = 52)	%	Total (N = 136)	%	χ or F	*p*-Value
Age:								
<35	37	44.0	17	32.7	54	39.7	χ^2^ = 1.7299	0.1884
≥35	47	56.0	35	67.3	82	60.3		
Total	84	100.0	52	100.0	136	100.0		
Age *	38.5; 36.5 (23–58)		40.1; 38 (28–68)		39.1; 37 (23–68)		F = 1.42	0.1770
Sex:								
Male	30	36.6	25	48.1	55	41.0	χ^2^ = 1.7366	0.1876
Female	52	63.4	27	51.9	79	59.0		
Total	82	100.0	52	100.0	134	100.0		
Missing	2		-		2			
Living alone:								
Yes	20	27.4	10	23.3	30	25.9	χ^2^ = 0.2421	0.6227
No	53	72.6	33	76.7	86	74.1		
Total	73	100.0	43	100.0	116	100.0		
Missing	11		9		20			
Worry about transmitting the virus to the people you live with:								
No/a little	16	30.8	17	56.7	33	40.2	χ^2^ = 5.3057	0.0213
A lot	36	69.2	13	43.3	49	59.8		
Total	52	100.0	30	100.0	82	100.0		
Missing	1		3		4			
Year of work experience								
	71; 15.9; 15 (0–36)		43; 10.2; 6 (0–41)		114; 13.7; 11 (0–41)		F = 1.45	0.1955
Missing	13		9		22			
Work area before the pandemic:								
ICU	49	62.0	38	76.0	87	67.4	χ^2^ = 2.7233	0.0989
Operating room/surgery/Other	30	38.0	12	24.0	42	32.6		
Total	79	100.0	50	100.0	129	100.0		
Missing	5		2		7			
Difficulties in adapting to the new work environment:								
Not at all	31	40.3	26	55.3	57	46.0	χ^2^ = 2.6649	0.1026
Yes	46	59.7	21	44.7	67	54.0		
Total	77	100.0	47	100.0	124	100.0		
Missing	7		5		12			
Compared to before the Covid emergency the patient number/ workload has:								
Increased	40	51.9	33	68.8	73	58.4	χ^2^ = 3.4359	0.0638
Equal/diminished	37	48.1	15	31.3	52	41.6		
Total	77	100.0	48	100.0	125	100.0		
Missing	7		4		11			
Compared to before the Covid emergency, how do you evaluate the relationship with your colleagues?								
Improved	41	55.4	30	60.0	71	57.3	χ^2^ = 0.2574	0.6119
Equal/Got worse	33	44.6	20	40.0	53	42.7		
Total	74	100.0	50	100.0	124	100.0		
Missing	10		2		12			
Compared to before the Covid emergency, there are opportunities to discuss important decisions in groups:								
More frequent	33	42.9	27	56.3	60	48.0	χ^2^ = 2.1248	0.1449
Equal/Less frequent	44	57.1	21	43.8	65	52.0		
Total	77	100.0	48	100.0	125	100.0		
Missing	7		4		11			
Do you feel protected when you work?								
No/a little bit	33	41.3	22	43.1	55	42.0	χ^2^ = 0.0455	0.8310
Yes	47	58.8	29	56.9	76	58.0		
Total	80	100.0	51	100.0	131	100.0		
Missing	4		1		5			

* Mean; Median (Range).

**Table 2 ijerph-18-06102-t002:** Results of logistic regression model (probability of burnout).

	OR ^1^	CI 95%	*p*-Value
Age			
<35 vs. ≥35	1.15	0.42–3.15	0.7851
Sex:			
Male vs. Female	0.86	0.32–2.33	0.7684
Profession			
Physicians vs. Nurses	1.19	0.45–3.11	0.7277
Living alone			
Yes vs. No	0.97	0.30–3.13	0.9619
Work area before the pandemic			
ICU vs. Operating room/Surgery/Other	3.02	1.06–8.58	0.0384
Difficulties in adapting to the new work environment:			
Yes vs. Not at all	1.55	0.59–4.10	0.3772
Compared to before the Covid-19 emergency, the patient number/workload has			
Equal/diminished vs. Increased	1.50	0.53–4.28	0.4443
Compared to before the Covid-19 emergency, how do you evaluate the relationship with your colleagues?			
Equal/Got worse vs. Improved	1.91	0.63–5.85	0.2555
Compared to before the Covid-19 emergency, there are opportunities to discuss decisions in groups			
Equal/Less frequent vs. More frequent	0.56	0.18–1.69	0.3003
Do you feel protected when you work?			
No/a little bit vs. Yes	1.35	0.49–3.69	0.5641
Anxiety			
Symptoms vs. Normal	2.39	0.81–7.05	0.1146
Depression			
Symptoms vs. Normal	4.88	1.54–15.48	0.0071
Resilience			
Moderate/High vs. Low or very low	0.76	0.20–3.00	0.7004
Insomnia			
Presence of symptoms of insomnia vs. No clinically significant insomnia	2.33	0.82–6.65	0.1129

^1^ Number of observations used = 136.

## Data Availability

The datasets analyzed during the current study are available from the corresponding author upon reasonable request.

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
