# Peer review of "Burnout in Intensive Care Unit Workers during the Second Wave of the COVID-19 Pandemic: A Single Center Cross-Sectional Italian Study"

_ijerph, 2021, doi:10.3390/ijerph18116102_

Round 1
Reviewer 1 Report
===========
International Journal of Environmental Research and Public Health
-----------------
Manuscript:
“Burnout in intensive care unit workers during the second wave 2 of the COVID-19 pandemic: A single center cross-sectional Italian study,” by Nino Stocchetti et al.
-----------------------------------------------------------------
The manuscript presents new results on distress, burnout syndrome and resilience in the ICU workers at Milano Fiera, Italy. These results provide additional evidence about enormous pressure on the frontline medical staff, effects on their mental health, psychological wellbeing, and resilience during the second wave of the COVID-19 outbreak. The authors confirmed previous finding on burnout levels and its components. These results are important for clinical practice. They can help (i) to identify groups of the ICU caregivers who are at a high risk, and (ii) to find factors and possible interventions to mitigate detrimental effects.
I have a few comments that I hope can help to make the manuscript stronger.
(1) Connections between Methods and remaining text can be strengthened by providing citations to the Methods where it is appropriate throughout the text and clearly indicating the method(s) used to obtain a particular result.
(2) The authors provided insufficient comparison of the observed detrimental effects and their components during the second wave of the COVID-19 outbreak with those before the pandemic, and with those during the first wave. This comparison would strengthen the manuscript substantially.
(3) The authors reported high rate of resilience (~82%) and burnout (~60%). Substantial difference in these rates indicates either incomplete questionnaire on Resilience or shared mediation effects. Could the authors provide comments on this aspect?
----------------------
MINOR COMMENTs:
----------------------
(4) Table 1: I would suggest writing each question as a title for in the related section, i.e. to use whole width of this Table but not the first column only. This will reduce its size and improve its readability, i.e. facilitate reading of the text in the horizontal but not in the vertical direction.
(5) Table 1: There are no descriptions and notes that explain meaning of chi2 and p-values. I would propose to introduce corresponding Notes and/or Comments and/or cite Methods with related information.
(6) Figure 1: I would recommend reversing the order of notations in all sub-plots and make them consistent with the color pattern presented on these plots. For instance, “burnout” region is depicted on the right side of the diagram in Fig.1a, but the color note was placed on the left side. It would be more natural to place the notations in the following order: blue color “no burnout”, green color “burnout”, and respectively in Figs.1b, 1c, and 1d. The same comments are related to Fig.2d.
(7) It would be informative to provide all figures with titles and descripts.
(8) I would suggest citing Figs.1a, 1b, 1c and 1d in appropriate context in a paragraph (lines 199-205) just above Figure 1. It will improve readability of the manuscript and will help the readers to connect related information and figure(s). The same comments are related to the entire section “Mental outcomes” (lines 199-235). I would suggest citing Tables and Figures when the authors discuss any results, i.e. when they provide numbers (percentage and p-value).
(9) There are dots in Fig.1d instead of “52.9%”.
(10) Does “IC 95%” in Supplementary Tables mean “CI 95%” == Confidence Interval?
(11) The authors provided description of the results from Supplementary Table A (Emotional Exhaustion), see lines 254-260), but they did not describe details of the results from Supplementary Tables B and C. Could the authors explain why they did not provide such a description, or could the authors include the description of the mentioned Supplementary Tables B and C?
Author Response
Dear reviewer,
Please find attached all the notes related to your comments.
Thanks for appreciating our work.
Best regards,
The authors

Reviewer 2 Report
In the present study, the authors investigated the prevalence of distress (defined as anxiety, depression and insomnia symptoms), burnout syndrome and resilience in healthcare workers during the COVID-19 pandemic in Lombardia (the most affected Italian region) and to detect potential factors associated with their psychological response. The topic was fascinating. However, the paper, in the current form, requires several corrections.
Introduction
This section should be implemented because it was not so clear, and it is not easy to understand the scope of the study. Many concepts are introduced without following a logical way. You should review this part and rewrite it.
Materials and Methods
1) The authors stated that the present is a cross-sectional web-based study. They used the hospital’s mailing list to send the online questionnaire. However, they did not report data about “how” they collected the data, avoided duplicates and preserved the anonymity of responders. For example, did you use survey monkey or google service? This section should be implemented.
2) I noted that the authors did not request information about years of work experience. This aspect is not of secondary importance, because a novice operator does not have the same experience in managing the same situations. In an emergency situation, years of experience could be considered as a possible bias, both in a positive and negative sense. If possible, authors should include this parameter in their analysis.
3) The authors, in the demographic questions, tried to obtain details on the “living conditions” of participants. However, they did not specify the meaning of “living condition”. Was this topic related to the family situation of the subject (i.e. single, divorced, married) or related to self-perception of quality of life or economic well-being? This aspect should be implemented and well analysed to avoid further bias in the statistical analysis.
3) For continuous variables, did the authors perform the test for normal distribution? In the paper, this was not stated. I suggest to the author to use test for normal distribution and to present in the proper manner the data.
4) Did the odd ratios calculate with a univariate logistic regression analysis?
5) Why did the authors chose 35 years-old as cut-off to dicotomise the studied population? The authors should provide details on this aspect.
Results
Unfortunately, this section was very confusing, difficult to read and interpret the data. Furthermore, the tables did not help the reader to understand the data. This section and tables should be completely rewritten.
Consequently, due to the severe difficulty in reading and understanding the data in the result section, I did not evaluate the discussion section.
In conclusion, the present manuscript requires a major revision.
Author Response
Dear reviewer,
Please find attached the notes to your comments
Best regards,
The authors.

Round 2
Reviewer 2 Report
Dear authors
I read your comments on my previous review report: "We regret that the reviewer did not have the tenacity, curiosity and willingness to conclude the reading we have voluntarily given ".
On this point, I want to clarify. As a journal reviewer, my role is to try to increase the understanding of your results. I want to remind you that "good data" poorly presented can invalidate the scientific role of work. I am sorry that my honesty on the matter has led you to judge me in a not so "positively" way.
Next time, I invite the authors to consider that the reviewer's work is done by specialists who try to help the authors present good work, highlighting the strengths and possible weaknesses of the paper. Aside from that, I reported my review below.
Your paper revision is well done. I noted only a few "stylistic" mistakes.
Introduction: Post-traumatic stress disorder should be in extensive form at first use. Please correct "PTSD" specifying the extensive form and then use abbreviation in the next part of the text. Moreover, I invite you to check for similar mistakes and correct.
Methods: You reported, in this section, that age presented continuous distribution. The authors should write this sentence in the results section.
Results: Data in table 1 are not well-formatted. I supposed that this was a problem related to the word software format used. In the present form, data were hard to understand. In table 2, the last line, "insomnia" reported incomplete data (OR, p-value).
When you reported results of univariate or multivariate logistic regression in the text, please insert p-value (i.e." With regard to gender, men reported lower levels of high EE than women (OR 0.46, 95% CI 0.22-0.94")).
Discussion: this section is well written, clear.
In conclusion: minor revision.
Author Response
Dear reviewer, Please see below the answers to the points that you've mentioned. Introduction: Post-traumatic stress disorder should be in extensive form at first use. Please correct "PTSD" specifying the extensive form and then use abbreviation in the next part of the text. Moreover, I invite you to check for similar mistakes and correct. --> thanks, we've amended that and also added some abbreviations also for the "HADS" (hospital anxiety and depression scale)Methods: You reported, in this section, that age presented continuous distribution. The authors should write this sentence in the results section. --> we've added this information in the results section.
Results: Data in table 1 are not well-formatted. I supposed that this was a problem related to the word software format used. In the present form, data were hard to understand. In table 2, the last line, "insomnia" reported incomplete data (OR, p-value). --> yes, it was depending on the IJERPH word template, but we've tried to find another solution to make it clearer.
When you reported results of univariate or multivariate logistic regression in the text, please insert p-value (i.e." With regard to gender, men reported lower levels of high EE than women (OR 0.46, 95% CI 0.22-0.94")). --> we've checked and the only sentence without the OR was "High levels of burnout (high EE/ low PA) were reported in those who didn’t feel protected when working" , we've added the OR for both EE and PA. We hope that the changes that we've made are in agreement with your recommendations and suggestions. Best regards and thanks, The authors.